# Revisiting the differential freezing nucleus spectra derived from drop freezing experiments; methods of calculation, applications and confidence limits

Gabor Vali

Deaprtment of Atmospheric Science, University of Wyoming, Laramie, Wyoming, USA

**Correspondence:** vali@uwyo.edu

Revised February 4, 2019

**Abstract.** The differential nucleus concentration defined in Vali (1971) is re-examined and methods are given for its applica-
tion. The purpose of this document is to facilitate the use of the differential spectra in describing the results of drop freezing, or
similar, experiments and to, thereby, provide additional insights into the significance of the measurements. The additive nature
of differential concentrations is used to show how the background contribution can be accounted for in the measurements. A
method is presented to evaluate the confidence limits of the spectra derived from given sets of measurements.

## 1  Introduction

Ice nucleation, more specifically freezing nucleation, remains a topic of interest in a variety of disciplines. Experiments with
multiple, externally identical, sample units have demonstrated the range of activities present in most samples, both for known
materials added to the water or for water derived from precipitation, lakes, rivers, or other sources. Freezing experiments are
important sources of information about ice nucleating particles (INPs) and hence are in fairly widespread use. This paper
addresses the calculation and utilization of the differential nucleus spectrum[1] derived from data obtained in drop freezing
experiments and denoted as $k(T)$. The closely related cumulative spectrum has been widely used already because of its direct
connection to the readily obtained fraction frozen. These functions were originally defined in Vali (1971; V71) and their link to
different forms, namely the differential and integral site density functions, is described in Vali (2014; V14). All these different
forms represent quantitative descriptions of the abundance and activity of ice nucleating particles (INPs) present in water
samples as functions of temperature. The abundance (concentration) is defined either with respect to the volume of water in
which the INPs are suspended or to the mass or total surface area of the INPs themselves. These functions are empirical results
that represent the most relevant characteristics (activity described in terms of the characteristic temperature) of the INPs based
on the singular model of freezing nucleation. This model is time-independent and is justified by the much greater influence of

---

[1]Strictly speaking the quantity of interest is the differential nucleus concentration. The differential spectrum is the graphical representation of the concen-
tration. However, it is convenient to refer to both as spectra.

temperature than of time in the activity of INPs. Justification for this manner of describing INP activity, as well as the degree to which time-dependence may alter the singular description, are presented in more detail in V14.

The spectra defined in the preceding paragraph are useful for quantitative definitions of activity as a function of temperature for given INPs, and to distinguish different INP populations by their activity. They also provide measures of ice formation in clouds, deduced from tests with precipitation samples. In the following, the differential spectrum is given most emphasis, partly because it is less well known, and more importantly because it is perhaps the most effective definition of INP activity in a sample. All impacts of INPs depend on temperature; the specific activity expected at some temperature, quantitatively expressed, is the information most relevant to the impact being studied[2]. Perhaps most important is the fundamental perspective that motivates these studies. We would like to have clearer understanding of the surface and kinetic factors that determine ice nucleation activity and of the temperature dependence of those factors. The abundance of nucleating sites of different activities (characteristic temperatures) for given substances is the key information which need to be explained in terms of structural and compositional features of the surfaces. This is the empirical input needed to formulating theories of ice nucleation.

There are many analogs in physics to the differential concentration information here discussed. The most prominent is perhaps the spectral intensity of light. More mundane is the population distribution by age group. In these examples, each segment of the spectrum, or age group can be directly observed and quantified. However, this is not the case in freezing experiments, because freezing of a drop at some temperature forecloses getting information about other potential INPs active at colder temperatures. These INPs not directly detectable have to be accounted for in order to get a meaningful result. Thus, it is necessary to obtain data with many drops in order to arrive at measures of the population at all temperatures. This problem is treated in the derivation of $k(T)$ in V71.

From and experimental perspective, quantitation of ice nucleating ability depends on a successful choice of the drop sizes and of the amount of suspended INPs. Because ice nucleating ability in general is a strong function of temperature, small drop volumes and low amounts of particle content result in freezing temperatures at low temperatures. On the contrary, with large drop volumes and high particle loading, most drops will freeze at roughly the same temperature. The range of usable drop volumes is often defined by the design of the apparatus, but, for laboratory preparations, particle concentration is controlled by the experimenter. For water samples obtained with indigenous INPs (rain, river water, etc.) particle concentrations can be altered by dilution and partial evaporation. The functions defined in the following section are useful only when the data to be analyzed describe a substantial spread of observed freezing temperatures.

---

[2]The dominant role of temperature in determining activity is dimmed somewhat by the fact that gradual cooling from above $0°$C is usually involved before reaching the specific temperature of activity. This introduces a combination of influences from the whole sequence of temperatures. Gradual cooling is the case for laboratory experiments with previously prepared samples and also in clouds if the majority of INPs get incorporated into cloud droplets before cooling to sub-zero temperatures. In some experiments and in some cloud situations, INPs enter into the water droplets (samples) at the supercooled temperature of interest, but in these cases observed freezing events may include effects often referred to as contact nucleation. This complication is set aside in this paper, so the nucleus spectra have to be viewed with that caveat in mind. This simplification is of relatively minor magnitude, as argued in Vali (2008) and in references quoted there.

Because the differential spectra are additive, i.e. represent the sum at each temperature of the contributions from all sources of the INPs in a given water sample, the differential spectra provide a way to correct for background noise in drop freezing experiments. This correction is detailed in Vali (2018) and in Section 6 of the paper. Another advantage of the differential spectrum is that confidence limits can be calculated for each point of the spectrum over the temperatures covered by the measurements. This is detailed in Section 7.

## 2  Definitions

The INP[3] spectra are derived from drop freezing experiments. The term drop freezing experiment is used here to represent the class of experiments in which freezing is observed with multiple subunits drawn from a sample of water containing dispersed ice nucleating particles (INPs). The experiments involve steady cooling of a number, $N_o$, of drops and the freezing temperature of each drop, $T_i$, is recorded. In practice, several runs with the same sample may be combined to accumulate a sufficiently large sample size $N_o$ for useful statistical validity of the results. Such a step, as practically all what is treated in this paper, assumes that the sample is stable, that is unaltered in any way during the time the measurements are performed.

The differential nucleus concentration, $k(T)$ is defined in Eq. (11) of V71 as

$$k(T) = -\frac{1}{X * \Delta T} * ln(1 - \frac{\Delta N}{N(T)})$$

(1)

where $T$ stands for temperature in °C, $N$ is the number of drops not frozen, $\Delta N$ is the number of freezing events observed between $T$ and $(T - \Delta T)$ i.e. drops for which $(T - \Delta T) < T_i < T$ and $X$ is the normalization to unit volume of water, unit mass or surface of INPs, or else, of the INPs. It is to be remembered that this expression is the result of considering that a freezing event in the interval $\Delta T$ is the result of a drop containing *at least* one INP active in that temperature interval (cf. V71). For relatively small $\Delta T$-values and for large $N$ this approximation to having a *single* INP per drop responsible for the observed freezing event is very good (and can be quantified from the properties of the Poisson distribution).

For experiments with adequate number of drops, the value of $\Delta N/N(T)$ is going to be small, so that an approximate expression is valid with negligible error, except for the lowest temperatures observed, when $N(T)$ also becomes small. The error in $k(T)$ (deviation from the exact equation Eq. 1) reaches 10% when $\Delta N/N(T)$ exceeds 0.2. This estimate is based on the fact that for a Poisson distribution the standard deviation is equal to the square root of the mean (cf. Ch. 9 in Blank (1980)). The approximate relationship is:

$$k(T) = \frac{1}{X * N(T)} * \frac{\Delta N}{\Delta T}; \quad \text{for} \quad \frac{\Delta N}{N(T)} \rightarrow 0.$$

(2)

---

[3]In all of the following the terminology given in Vali et al. (2015; V15) is followed

The cumulative concentration, the integral of $k(T)$ over temperature, is given by Eq. (13) in V71 as:

$$K(T) = \frac{1}{X} * [lnN_o - lnN(T)] \tag{3}$$

which can be re-written in terms of the fraction of drops frozen $f(T)$ as

$$K(T) = -\frac{1}{X} * ln[1 - f(T)] \tag{4}$$

Because $f(T)$ is readily obtained in most experiments, this direct link to $K(T)$ is used in a number of publication (e.g. DeMott et al., 2017; Hader et al, 2014; Häusler et al. 2018; Harrison et al. 2018; Kumar et al. 2014; Paramonov et al, 2018; Tarn et al, 2018, Whale et al., 2015 ) to represent the results in terms of $K(T)$.

A third alternative to obtaining $K(T)$ is to do a numerical integration of $k(T)$, remembering that the $k(T)$ values here are at discreet $T$ values, not a function:

$$K(T) == \sum_{0}^{T} k(T) \cdot \Delta T. \tag{5}$$

For normalization of $k(T)$ or $K(T)$ to unit volume of the water $X = V$ where $V$ is the volume of the drops, assuming drops of uniform sizes. For normalization to unit surface area of material dispersed in the drops $X = A$ with $A$ denoting the average surface area of particles in each drop. In this case, many authors replace $K(T)$ by $n_s(T)$ where $n_s$ stands for the site density. See Section 8 for further discussion of the determination of active site density.

Mention has been made already that sample stability is assumed for valid representations of nucleating activity in any quantitative way. Since most INPs are insoluble solid materials they can be considered stable. Many different potential site configurations, such as crystal steps, dislocations, cracks, voids, inclusions, adsorbed substances are likely to be stable. However, since ice nucleation takes place on the substrate surface, stability of the surface is required and that is much more diffucult to be assured of. The stability requirement is clearly not fulfilled by samples such as cellulose because they undergo changes

when introduced into water. In general, the applicability of active site density may not be known *a priori*, but can be assessed by testing for consistency with different particle loadings, treatments or other methods.

A great advantage of quantitating ice nucleating ability in terms of the spectra defined here is the simplicity of these quantities. No assumptions are needed about intrinsic particle properties, as for example contact angle, and neither are the results interpreted in terms of quantities not readily determined independently. While presentation of empirical results as counts of

INPs may seem overly simple, the spectra are good measures of expected ice nucleation in the water samples tested and, for

prepared suspensions of known materials, $k(T)$ and $K(T)$ can readily be used as the basis of refinements in terms of different models of material properties and site configurations. The first step in that direction is the active site density description discussed in Section 8.

## 3  Sample data

Data from an experiment with a $\mathrm{Snomax}^{\mathrm{TM}}$ sample is used here[4] for demonstrating the manner of calculating the differential concentration. Observed freezing temperatures for 507 drops are listed in Table 1. The observations were made with steady cooling of the drops. Freezing events spread over the temperature range from near $-4°$C to near $-35°$C. Freezing events are most frequent in two temperature regions, one near $-8°$C and the other at the lowest temperatures. As can be seen, some temperature values occur more than once due to the finite resolution of the detection and recording system used. These
characteristics of this data make it useful to demonstrate various points about the calculations.

## 4  Choice of temperature interval

The main decision in applying either Eq. (1) or (2) to experimental results is what numerical values to use for $\Delta T$, taking into account constrains arising from the resolution of the temperature measurements and from finite sample sizes. While all other quantities in Eqs. (1) to (3) are directly measured, $\Delta T$ is not an empirical value but is one chosen in analysis for desirable
representation of the observations. For the assumptions involved in the derivation of $k(T)$, as described in V71, infinitesimally small intervals $\delta T$ should be applied, but this would necessitate infinite, or very large, sample sizes $N_o$ in order to avoid a large number of intervals without any events. Thus a finite $\Delta T$ is required. It will be argued that a uniform $\Delta T$ over the entire temperature range of an experiment is the simplest and most effective choice. The choice is made, principally, on the basis of sample size (number of drops in the experiment) and not based on instrumental variables, such as the recording interval of
freezing events.

One possible solution for calculating $k(T)$ with high resolution would be to use $\Delta N = 1$ and with the temperature intervals between freezing individual events as $\Delta T$. This would yield as many points on the spectrum plot as the number of drops. However, this approach would have variable $\Delta T$-values which in turn leads to variations in the calculated $k(T)$ values. The magnitude of each point would depend on the temperature interval between successive freezing events. A given freezing event
would correspond to a $k(T)$ value whose magnitude is changed depending on the previous freezing event in the sample. In effect, the quantitive significance of the results would be negated. To see this for the Snomax data, the temperature gaps, the differences between the freezing temperatures for successive events are shown in Fig. 1. Each point corresponds to one drop and is plotted at the freezing temperature of that drop. The large number of points at zero gap size indicate coincidences in the recorded temperatures for several drops due to the finite resolution of the recording system. Another grouping of points
just below 0.1 is due to the temperature change during the time intervals with which the number of frozen drops was recorded.

---

[4]These data are from work described in Polen et al. (2018) and are used here with kind permission by Dr. Ryan Sullivan of Carnegie Mellon University.

**Table 1.** Observed freezing temperatures for 507 drops of a sample of Snomax$^{\text{TM}}$ dispersed in purified water. Freezing temperatures are listed in decreasing order. Multiple values are due to time steps of the detection system used. These data are from work described in Polen et al. (2018).

| | | | | | | | | | | | | |
|---|---|---|---|---|---|---|---|---|---|---|---|---|
| -4.42 | -6.34 | -6.63 | -6.71 | -6.79 | -6.84 | -6.84 | -6.92 | -6.92 | -6.92 | -7.01 | -7.01 | -7.01 |
| -7.01 | -7.01 | -7.05 | -7.14 | -7.14 | -7.14 | -7.14 | -7.14 | -7.14 | -7.14 | -7.21 | -7.21 | -7.21 |
| -7.21 | -7.29 | -7.29 | -7.29 | -7.29 | -7.29 | -7.34 | -7.34 | -7.34 | -7.43 | -7.43 | -7.43 | -7.43 |
| -7.50 | -7.50 | -7.50 | -7.50 | -7.57 | -7.57 | -7.57 | -7.57 | -7.57 | -7.57 | -7.57 | -7.57 | -7.57 |
| -7.57 | -7.57 | -7.63 | -7.63 | -7.63 | -7.63 | -7.63 | -7.63 | -7.63 | -7.63 | -7.71 | -7.71 | -7.71 |
| -7.71 | -7.71 | -7.71 | -7.71 | -7.71 | -7.71 | -7.71 | -7.79 | -7.79 | -7.79 | -7.79 | -7.79 | -7.79 |
| -7.79 | -7.86 | -7.86 | -7.86 | -7.86 | -7.86 | -7.86 | -7.93 | -7.93 | -7.93 | -7.93 | -7.93 | -7.93 |
| -7.93 | -7.93 | -7.98 | -7.98 | -7.98 | -7.98 | -8.05 | -8.05 | -8.05 | -8.05 | -8.05 | -8.05 | -8.05 |
| -8.05 | -8.05 | -8.05 | -8.11 | -8.11 | -8.11 | -8.21 | -8.21 | -8.21 | -8.21 | -8.21 | -8.21 | -8.21 |
| -8.27 | -8.27 | -8.27 | -8.27 | -8.27 | -8.27 | -8.27 | -8.34 | -8.34 | -8.40 | -8.40 | -8.40 | -8.40 |
| -8.40 | -8.40 | -8.50 | -8.50 | -8.55 | -8.55 | -8.55 | -8.55 | -8.55 | -8.55 | -8.63 | -8.63 | -8.70 |
| -8.70 | -8.77 | -8.77 | -8.77 | -8.84 | -8.84 | -8.84 | -8.84 | -8.92 | -8.99 | -8.99 | -8.99 | -8.99 |
| -9.06 | -9.06 | -9.06 | -9.06 | -9.06 | -9.12 | -9.21 | -9.21 | -9.26 | -9.35 | -9.50 | -9.55 | -9.55 |
| -9.71 | -9.79 | -9.93 | -10.00 | -10.00 | -10.08 | -10.13 | -10.29 | -10.34 | -10.57 | -10.57 | -10.64 | -10.71 |
| -11.29 | -11.29 | -11.36 | -11.94 | -11.94 | -11.94 | -12.02 | -12.16 | -12.69 | -12.69 | -12.92 | -13.28 | -13.48 |
| -13.56 | -13.99 | -14.42 | -14.94 | -15.30 | -15.67 | -16.03 | -16.82 | -16.82 | -17.19 | -17.32 | -17.54 | -19.30 |
| -20.40 | -20.85 | -21.13 | -21.13 | -21.87 | -22.66 | -23.73 | -23.73 | -24.12 | -24.17 | -24.26 | -25.06 | -25.34 |
| -25.42 | -25.77 | -25.84 | -26.07 | -26.29 | -26.36 | -26.51 | -26.56 | -26.65 | -26.93 | -27.07 | -27.07 | -27.30 |
| -27.65 | -27.81 | -27.87 | -27.94 | -28.08 | -28.31 | -28.36 | -28.47 | -28.52 | -28.60 | -28.60 | -28.68 | -28.80 |
| -28.89 | -28.89 | -29.04 | -29.16 | -29.25 | -29.31 | -29.46 | -29.46 | -29.55 | -29.69 | -29.91 | -30.05 | -30.05 |
| -30.21 | -30.48 | -30.48 | -30.48 | -30.70 | -30.78 | -30.78 | -30.85 | -30.93 | -31.00 | -31.06 | -31.16 | -31.16 |
| -31.32 | -31.32 | -31.32 | -31.32 | -31.41 | -31.56 | -31.63 | -31.77 | -31.77 | -31.83 | -31.92 | -31.92 | -31.92 |
| -31.97 | -32.22 | -32.22 | -32.28 | -32.28 | -32.28 | -32.33 | -32.42 | -32.49 | -32.49 | -32.64 | -32.64 | -32.70 |
| -32.78 | -32.86 | -32.86 | -32.86 | -32.86 | -32.94 | -32.94 | -32.94 | -33.00 | -33.00 | -33.00 | -33.06 | -33.06 |
| -33.14 | -33.14 | -33.23 | -33.23 | -33.29 | -33.29 | -33.29 | -33.35 | -33.35 | -33.35 | -33.43 | -33.43 | -33.43 |
| -33.43 | -33.43 | -33.43 | -33.49 | -33.49 | -33.49 | -33.49 | -33.49 | -33.49 | -33.59 | -33.59 | -33.59 | -33.59 |
| -33.59 | -33.59 | -33.59 | -33.59 | -33.59 | -33.59 | -33.65 | -33.65 | -33.65 | -33.65 | -33.65 | -33.65 | -33.65 |
| -33.65 | -33.65 | -33.65 | -33.71 | -33.71 | -33.71 | -33.71 | -33.71 | -33.71 | -33.71 | -33.71 | -33.79 | -33.79 |
| -33.79 | -33.79 | -33.79 | -33.79 | -33.79 | -33.79 | -33.79 | -33.79 | -33.79 | -33.79 | -33.79 | -33.79 | -33.79 |
| -33.79 | -33.86 | -33.86 | -33.86 | -33.86 | -33.86 | -33.86 | -33.86 | -33.86 | -33.86 | -33.86 | -33.86 | -33.86 |
| -33.86 | -33.86 | -33.86 | -33.86 | -33.92 | -33.92 | -33.92 | -33.92 | -33.92 | -33.92 | -33.92 | -33.92 | -33.92 |
| -33.92 | -33.92 | -33.92 | -33.92 | -33.92 | -33.92 | -33.92 | -33.92 | -33.92 | -33.92 | -33.92 | -33.92 | -33.92 |
| -33.92 | -33.92 | -33.92 | -33.92 | -33.92 | -33.92 | -33.92 | -34.01 | -34.01 | -34.01 | -34.01 | -34.01 | -34.01 |
| -34.01 | -34.01 | -34.01 | -34.01 | -34.01 | -34.01 | -34.01 | -34.01 | -34.01 | -34.01 | -34.01 | -34.01 | -34.01 |
| -34.01 | -34.01 | -34.01 | -34.07 | -34.07 | -34.07 | -34.07 | -34.07 | -34.07 | -34.07 | -34.07 | -34.07 | -34.07 |
| -34.07 | -34.07 | -34.07 | -34.07 | -34.07 | -34.07 | -34.07 | -34.07 | -34.07 | -34.07 | -34.13 | -34.13 | -34.13 |
| -34.13 | -34.13 | -34.13 | -34.13 | -34.13 | -34.13 | -34.13 | -34.13 | -34.13 | -34.13 | -34.13 | -34.23 | -34.23 |
| -34.23 | -34.23 | -34.23 | -34.23 | -34.23 | -34.23 | -34.23 | -34.23 | -34.23 | -34.23 | -34.23 | -34.23 | -34.23 |

Both the zeroes and these minimum non-zero values are most numerous near $-8°$C and near $-33°$C where there are high

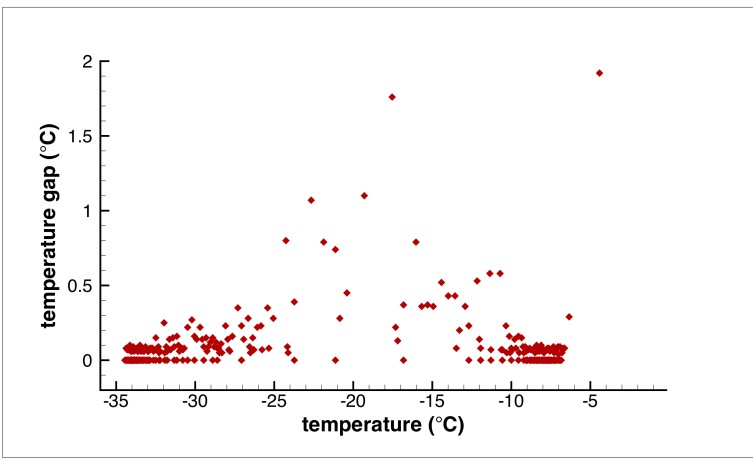

**Figure 1.** Temperature gaps between successive freezing events in the data given in Table 1. Fewer events in the middle range of temperatures produce fewer and larger gaps.

numbers of freezing occurrences. Larger gaps become more frequent in the temperature range between the two groups due to the sparsity of freezing events. These large and irregular gaps would scramble the $k(T)$ values.

Conversely, using a constant value across the range of temperatures covered by the data assures that all points are on the same scale. If the observed freezing temperatures are close to each other varying the interval width would be compensated by the inclusion of more or fewer events, so the results would be acceptable, but there is no practical reason for doing that. So, it is recommended to select a suitable value for $\Delta T$ and use it for the whole data set.

In the majority of experiments, $T_i$ is irregularly distributed over the range of all freezing events for a given sample. Thus, if $\Delta T$ is chosen too small there will be intervals with zeros and ones only. That would result in an almost meaningless representation of the results as $k(T)$ would also consist of zeros and a uniform small value. The density of points along the $T$ axis would show some pattern but only in a qualitative way. The value chosen for $\Delta T$ is a compromise between what's ideal and what's practical. The latter perspective of course involves judgements over several factors. Most importantly, these factors are the sample size and associated statistical validity, the precision with which $T_i$-values are determined, and the detail in the final spectrum that is believed to hold meaningful information. In view of these conflicting influences, there is no single recipe for setting $\Delta T$, but the variations that result in the specific choice do not diminish the objective value of the derived k(T) spectrum if normalized to unit temperature interval.

For the sake of simplicity and generality, equal drop volumes are assumed in the calculations here, $X$ is set to unity, and the differential concentrations are presented with units of $°C^{-1}$. Depending on the choice for $X$, (drop volume, particle surface area per drop, mass of particles per drop) the units of $k(T)$ will be different, such as, for example $°C^{-1}cm^{-3}$, or $°C^{-1}\mu m^{-2}$, or $°C^{-1}g^{-1}$.

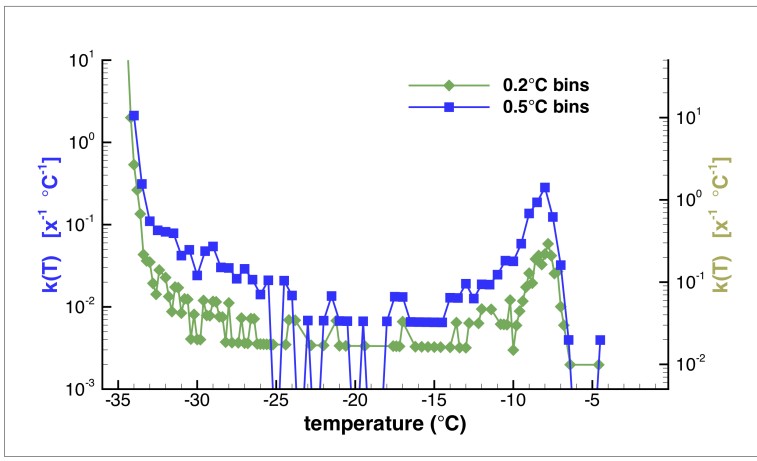

**Figure 2.** Plots of $k(T)$ for 0.2°C and 0.5°C bin sizes for the data from Table 1. The right-hand scale is shifted down slightly to allow the two plots to be clearly seen. Zero values are indicated for the 0.5°C graph with values below the range covered by the ordinate. The ordinate values are for $X = 1$ of unspecified dimension, and thus the units are given as $[\mathrm{x}^{-1}{}^{\circ}\mathrm{C}^{-1}]$.

To illustrate the impacts of the choice of $\Delta T$, Fig.2 shows the spectra for the Snomax sample with two different values. The data shown in Table 1 was binned using $\Delta T = 0.2$°C and $\Delta T = 0.5$°C. For $\Delta T = 0.2$°C there are 51 empty bins (zeroes) between $-6$ and $-34$°C. For $\Delta T = 0.5$°C there are only 8 zeroes in the same temperature range. Eq. (2) was then used to obtain $k(T)$. Plots of $k(T)$ shown in Fig. 2 differ, principally, in the degree of noisiness of the data points. Because of the large range of values covered, plots of $k(T)$ almost always use logarithmic ordinate scale. This eliminates the possibility to include zero values, and special steps need to be taken for the plots to show these values. For one of the plots in Fig.2 the zeroes were replaced by a low value well below the range covered by actual data in order to indicate the presence of the zero values. Without this, the presence of zeroes, or empty bins, is seen as gaps between points, and as horizontal lines. This matters in judging the significance of the points surrounding the zeroes. Clearly, the dip in $k(T)$ between -26°C and -17°C is perceived to be much deeper when the zeroes are indicated.

## 5   Calculation of $k(T)$ and $K(T)$.

Once the interval width has been decided, calculation of the differential concentration is a straightforward matter, resulting in a value of $k(T)$ for each temperature interval. The cumulative concentration is then also calculated for the same temperatures if it is done by summation of the differential values. This is not a requirement; the cumulative spectrum can be also calculated without binning of the data and for as many temperatures as wanted.

Based on the comparison presented in Fig. 2 and in the text associated with it, calculations for the Snomax sample are processed here with $\Delta T = 0.5$°C. The result of that binning of $T_i$-values is shown in Fig. 3 as a histogram. After binning, values of $N(T)$ were calculated by stepwise addition of the $\Delta N$ values from the lowest to the highest temperature, ending up

**Table 2.** Differential and cumulative spectra for the Snomax sample with 0.5°C intervals, as discussed in Section 5.

| [1] temperature interval center $T$ | [2] number of events in interval $\Delta N$ | [3] number unfrozen at beginning of interval $N$ | [4] number frozen at end of interval $N_f$ | [5] fraction frozen at end of interval $f(T)$ | [6] differential per °C $k(T)$ | [7] cumulative at end of interval $K(T)$ |
|---|---|---|---|---|---|---|
| -3.75 | 0 | 507 | 0 | 0.000 | 0.000 | 0.000 |
| -4.25 | 1 | 507 | 1 | 0.002 | 0.004 | 0.002 |
| -4.75 | 0 | 506 | 1 | 0.002 | 0.000 | 0.002 |
| -5.25 | 0 | 506 | 1 | 0.002 | 0.000 | 0.002 |
| -5.75 | 0 | 506 | 1 | 0.002 | 0.000 | 0.002 |
| -6.25 | 1 | 506 | 2 | 0.004 | 0.004 | 0.004 |
| -6.75 | 8 | 505 | 10 | 0.020 | 0.032 | 0.020 |
| -7.25 | 29 | 497 | 39 | 0.077 | 0.120 | 0.080 |
| -7.75 | 58 | 468 | 97 | 0.191 | 0.265 | 0.212 |
| -8.25 | 35 | 410 | 132 | 0.260 | 0.178 | 0.302 |
| -8.75 | 24 | 375 | 156 | 0.308 | 0.132 | 0.368 |
| -9.25 | 10 | 351 | 166 | 0.327 | 0.058 | 0.397 |
| -9.75 | 6 | 341 | 172 | 0.339 | 0.036 | 0.414 |
| -10.25 | 6 | 335 | 178 | 0.351 | 0.036 | 0.432 |
| -10.75 | 4 | 329 | 182 | 0.359 | 0.024 | 0.445 |
| -11.25 | 3 | 325 | 185 | 0.365 | 0.019 | 0.454 |
| -11.75 | 3 | 322 | 188 | 0.371 | 0.019 | 0.463 |
| -12.25 | 2 | 319 | 190 | 0.375 | 0.013 | 0.470 |
| ..... | ..... | ..... | ..... | ..... | ..... | |
| ..... | ..... | ..... | ..... | ..... | ..... | |
| -28.75 | 7 | 265 | 249 | 0.491 | 0.054 | 0.676 |
| -29.25 | 6 | 258 | 255 | 0.503 | 0.047 | 0.699 |
| -29.75 | 3 | 252 | 258 | 0.509 | 0.024 | 0.711 |
| -30.25 | 6 | 249 | 264 | 0.521 | 0.049 | 0.735 |
| -30.75 | 5 | 243 | 269 | 0.531 | 0.042 | 0.756 |
| -31.25 | 9 | 238 | 278 | 0.548 | 0.077 | 0.795 |
| -31.75 | 9 | 229 | 287 | 0.566 | 0.080 | 0.835 |
| -32.25 | 9 | 220 | 296 | 0.584 | 0.084 | 0.877 |
| -32.75 | 11 | 211 | 307 | 0.606 | 0.107 | 0.930 |
| -33.25 | 27 | 200 | 334 | 0.659 | 0.290 | 1.075 |
| -33.75 | 89 | 173 | 423 | 0.834 | 1.445 | 1.798 |
| -34.25 | 84 | 84 | 507 | 1.000 | 0.000 | 1.798 |
| -34.75 | 0 | 0 | 507 | 1.000 | 0.000 | 1.798 |
| -35.25 | 0 | 0 | 507 | 1.000 | 0.000 | 1.798 |
| -35.75 | 0 | 0 | 507 | 1.000 | 0.000 | 1.798 |

with $N_o$ for the first interval with non-zero $\Delta N$. Doing the accumulation of $\Delta N$ from lowest to highest temperature produces

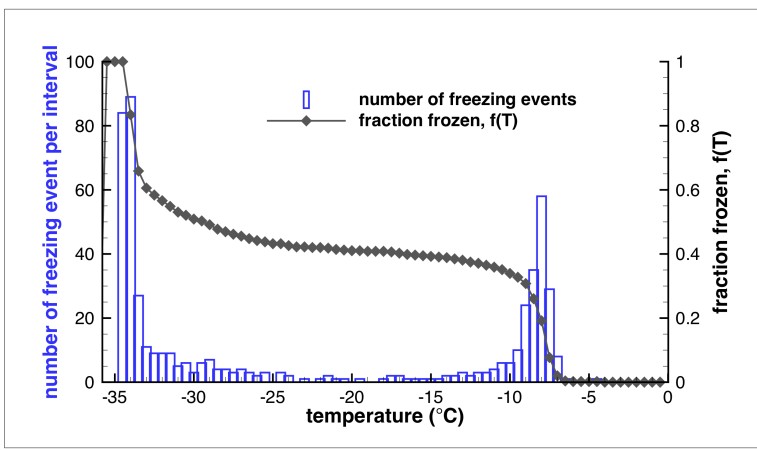

**Figure 3.** Histogram of freezing temperatures and a plot of the fraction of drops frozen for the data from Table 1 (Snowmax suspension).

$N$ values at the upper end (warmer temperature) of each interval. The fraction frozen expressed with respect to the lower end (colder temperature) of the interval is obtained as:

$$f(T) = 1 - \frac{N(T) - \Delta N}{N_o}.$$ (6)

The differential concentration was calculated from Eq. 1 and the cumulative from Eq. 3. Results are given in Table 2. The table is given from highest to lowest temperature to make it match the way the data are obtained in the experiment with gradual cooling. The temperature in the first column is the mid-point of the interval over which the data were evaluated. As indicated in the preceding paragraph, columns [4], [5], and [7] are shifted by one line with respect to the others in order that they refer to the low end of the temperature interval. These distinctions of interval mid-point, high and low end are somewhat unnecessary considering the magnitude of the interval width but are included here to avoid misinterpretation of the tabulated data. It is also worth noting that at the initial part of the table, the cumulative concentration is smaller in magnitude than the differential because the differential is normalized to °C intervals, making the values, for $\Delta T = 0.5$°C used in this example, double of the value without that normalization.

Plots of the differential and cumulative spectra are given in Fig. **??**. In this graph, zero values are skipped over for giving the graph a less cluttered appearance. By using the same ordinate for both plots, the cumulative curve starts lower than the differential, as explained above. Normalization to per unit volume of the drops or to site density is a matter of applying the relevant multiplier to the ordinate values. In this example, and in most of this paper, plots of the spectra are shown with individual points for each temperature interval. In some cases, it might be desirable to fit algebraic equations to the data.

The effectiveness of transmitting the results of analyses such as this, as mentioned, depends on the numerous factors already discussed. From a purely data-processing perspective, the spectrum with lower resolution is better because it has fewer zero

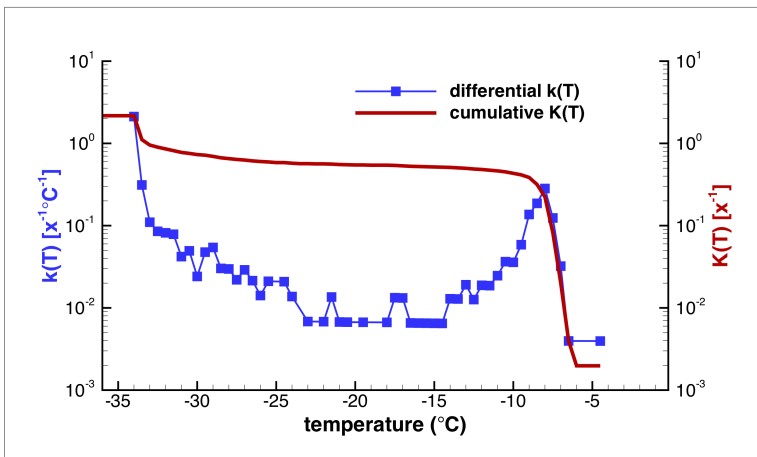

**Figure 4.** Differential and cumulative spectra for data discussed in Section 5 and displayed in different form in Figs. 2 and 3. Zeros in the differential spectrum are seen in this plots by larger gaps between adjacent points. The left and right ordinate scales are identical. As mentioned in the text, the cumulative curve starts at a lower value than the differential because of the differential is expressed with reference to full degree intervals.

values. No claim is made that the $\Delta T = 0.5°$C choice is optimal. The resulting $k(T)$ spectrum still has considerable fluctuations in the middle portion of the temperature range. On the other hand, the main peak is well resolved, as is its asymmetric shape. There are many additional steps that can be considered for smoothing the data, either at the $\Delta N$ level or in $k(T)$.

From the point of view of showing what kind of INPs were contained in the sample, all the graphs clearly indicate peaks
in activity near $-8°$C and near $-33°$C. The first peak is of the greater interest because it is due to the INPs added to the sample, while the low-temperature activity is due to the background related to the supporting surface of the drops and to impurities in the water used to suspend the active INPs. As a minor detail, it may be noted that he $-8°$C peak has a broader tail toward colder temperatures. This features is clearly seen in both of the graphs. Even finer details of the peak can be seen if the data are processed at higher resolution but very little significance can be attached to such details in light of the sample
size, the temperature precision of the measurements and other instrumental factors. Nonetheless, it is important to note that the differential spectra can resolve distinct peaks and thus can provide the type of acute description of INP activity that is needed in many studies.

## 6 Background correction

The differential concentration in a sample with various sources of INPs can be assumed to be the sum of the concentrations
due to each of the sources. This assumption of additive behavior is likely to hold for many cases and would be incorrect only if, for some reason, interactions are expected between INPs from the different sources. The most relevant example of additive

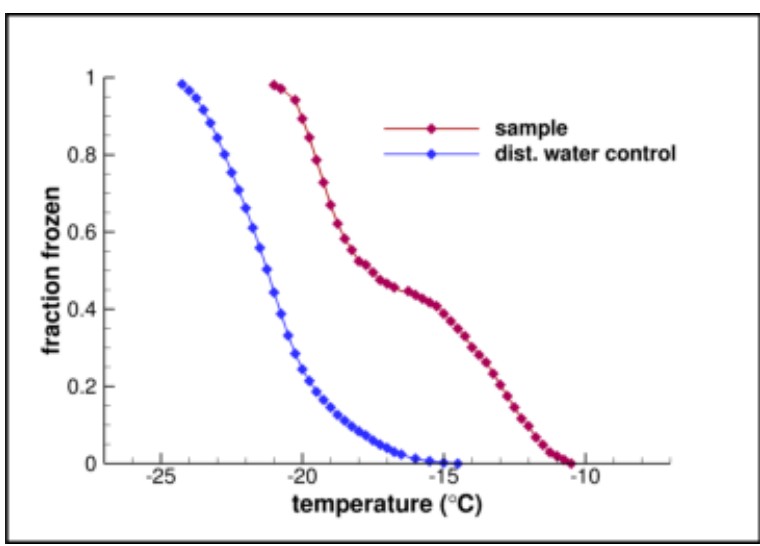

**Figure 5.** Observed fractions of droplets frozen for the soil sample and for the distilled water control, as described in Section 6. Data are from a single run with 103 drops of 0.01 cm$^3$ volume.

behavior, applicable to essentially all experiments with laboratory preparations, is the addition of the background activity to that of the material to be tested. The water used to prepare suspensions of INPs is never totally free of INPs, and there is potential for further contributions to the 'background' by the components of the apparatus used in the experiment. While extreme care is taken in most cases to minimize the background, it is always present to greater or lesser extent. Determination of the background is accomplished with control experiments.

The usefulness of a quantitative assessment of the background activity is demonstrated with the following example[5]. A suspension of soil particles in distilled water, and control measurements of the distilled water, yielded the fraction frozen curves in Fig. 5. From these graphs it would appear that the soil sample data are not reliable much below about −18°C because of the appreciable level of activity in the control. When the differential spectra are computed and the control is subtracted from the $k(T)$-values for the sample, the resulting plot shown in Fig. 6 reveals that only in a narrow region near −17°C is the contribution from the distilled water comparable to the INP activity in the soil. Thus, the INP activity in the soil sample below −18°C can be judged in a more objective fashion. Just considering this result, it would not be baseless to conclude that the soil sample contained two types of INPs, those producing the peak centered on −13°C and those giving rise to high numbers of INP below −18°C. In practice, further tests with different amounts of soil in suspension would be useful to judge that conclusion.

[5]This is the same example as was used in Vali (2018)

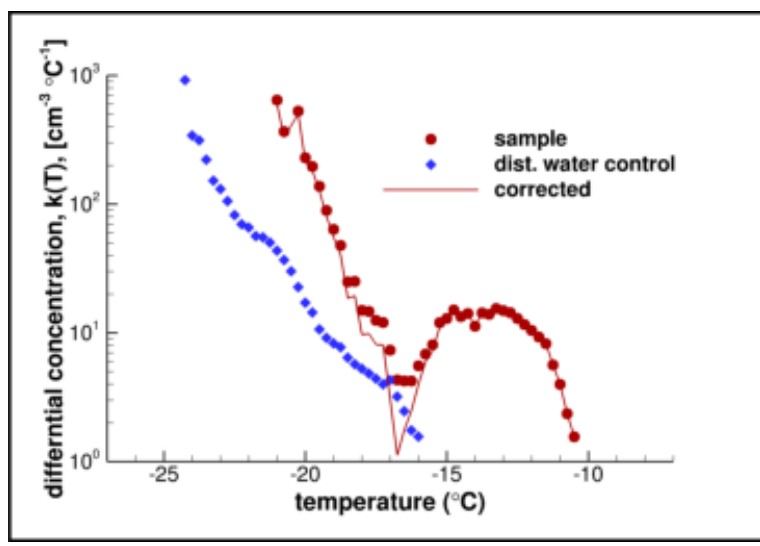

**Figure 6.** Differential spectra for the same data as shown in Fig. 5. Circle symbols are for the soil sample, diamond symbols are for the control (blue). The spectra for the soil sample after correction for the distilled water background is shown with a line. The magnitude of the correction is relatively minor in this case except in the temperature region between about $-14°$C and $-18°$C.

## 7 Confidence intervals

Several sources of error contribute to determining the confidence limits or uncertainty ranges of results derived from drop freezing experiments. Temperature accuracy is a minor contribution in most cases. Acuity of the detection of freezing is a larger concern. These and other error sources need to be evaluated specifically for each experimental setup. A general and demanding

problem is the evaluation of the statistical validity of results. That uncertainty, arising from sample sizes, is of special concern because of the usually large temperature range of the observations, and the consequent small number of freezing events at each temperature. Uncertainty ranges specific to each temperature can be evaluated using the $k(T)$ spectra, as described in the following.

    Even with identical drop volumes and with all drops produced from the same bulk suspension, considerable spreads in

freezing temperatures are usually observed. As discussed earlier, variations in freezing temperatures are associated with specific differences in INPs so that the variations in freezing temperatures indicate a non-random distribution of the INPs of different activities in the drops. Hence, basic statistical methods are not applicable to estimating the confidence interval of the $k(T)$ or $K(T)$ spectra derived to characterize the INP content. In the absence of many repetitions of the experiments to determine variability, Monte Carlo simulations provide a possible solution. In V71, such simulations were applied to show how the spread

in $k(T)$ spectra is reduced by increasing sample size. Monte Carlo methods of slightly different configurations were also used in Wright and Petters (2013) and in Harrison et al. (2016).

The differential concentration provides a convenient basis for simulations because values of $k(T)$ for given temperatures are independent of the values at other temperatures. Use of the cumulative concentration derived from the fraction frozen would be less transparent. The simplest basis for simulations is the number of freezing events observed in each temperature interval, $\Delta N(T)$. Random variability expected about those values is the measure sought in the simulation. This can be viewed as if a new set of drops were taken each time from the same bulk sample, or a new set of particles were dispersed into the volume each time, and then a freezing run performed. Simulation allows as many of these runs to be done as needed to reach a good estimate of the variability.

The simulation is relatively simple. The number of events in any given temperature interval can be expected to follow a Poisson distribution on repeated testing. This probability distribution fits the situation because the number of events per interval is discrete, independent of other intervals, and the observed numbers can serve as the assumed true values. Hence, taking the observed values of $\Delta N(T)$ as the expectation values $\lambda(T)$ and generating a large number, say $p$, Poisson-distributed numbers for each temperature interval provides independent virtual realizations of the experiment. The mean value of the $\Delta N_i$ ... $\Delta N_p$ numbers in each interval will equal $\lambda$ for that interval, and the standard deviation will be $\lambda^{0.5}$. However, the Poisson distributions include zeros even for mean values greater than zero. The chance of this reduces as the mean increases; the number of zero values is $e^{-\lambda}$.

For a first demonstration of the simulation, a data set with a modest number of 106 drops is used here. Measured numbers of freeing events for $\Delta T = 0.5°C$ intervals and the calculated values of $k(T)$ are given in Table 3. As can be seen, the number of events per interval is small, and would contain many zeroes using a smaller $\Delta T$. Values in the second column were taken as $\lambda$ and 100 new sets of $\Delta N_i$-values generated using a Poisson distributed random number generator in IDL (Harris Geospatial Solutions, Inc.). From those 100 new sets of values, 100 new $N(T)$-values were derived and $k(T)$ calculated using Eq. 1. The simulation results can be used in many different ways to represent the resulting uncertainties in the presentations of the empirical results. The scatter in $k(T)$ values is an immediate way to show the results. Cumulative spectra $K(T)$ can also be obtained, as can standard deviations, or other measures.

Simulated results in terms of $k(T)$ are shown in Fig. 7. At a few places above the temperature axis, the number of zero values that occurred in the simulation for that interval are indicated. In this approach, the total number $N_o$ for any given run is not constrained to $\sum \lambda$; the actual number among the 100 simulated sets varied by 10%. This variation alters the simulated $k(T)$ values at the low end of the temperature range to some degree but is insignificant at the high end. There seem to be little reason to go to that extent or refinement, but the problem could be eliminated by adjusting $\lambda$ for lower temperatures for each choice of $\Delta N_i$ in successive steps. One point of assurance on this score is that the 50-percentile of the simulated $k(T)$ points is only 3% off from those shown in Table 3.

The spread of 10 to 90% of values at each interval are shown in Fig. 8. This example shows roughly a factor of four spread in $k(T)$ over the whole range of temperatures; worse for those points with low $k(T)$ and hence also having zero values potentially

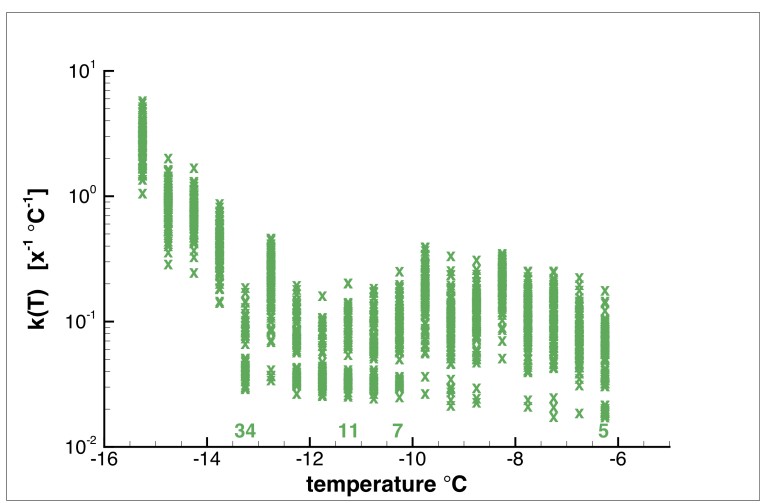

**Figure 7.** Calculated $k(T)$ values for 100 iterations of random assignments of $\Delta N$ from a Poisson distribution with the $\lambda$ values shown in Table 3 for each interval. Numbers above the abscissa indicate the number of zero values in the simulation for selected temperatures.

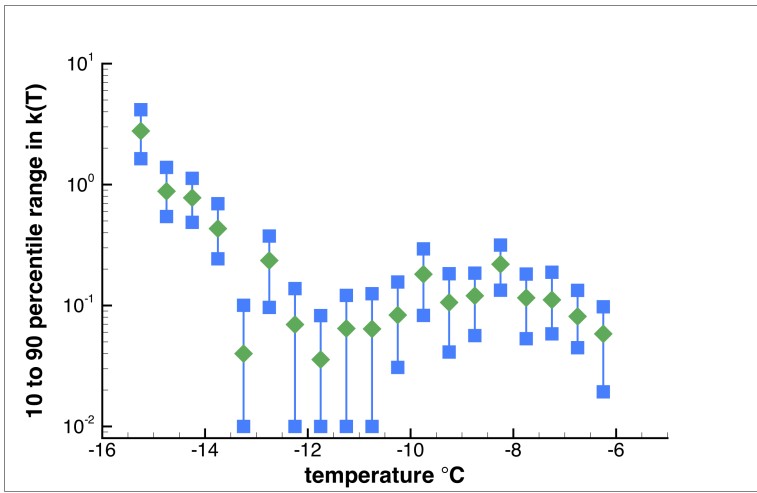

**Figure 8.** The 10 to 90 percentile range of $k(T)$ for the results shown in Fig. 7. The green diamonds show the values of $k(T)$ from the right-hand column of Table 3 for the observed sequence of freezing events. Points just above the abscissa are actually zero values.

**Table 3.** Observed freezing data used as input to the Monte Carlo simulation described in Section 7.

| temperature $T$ | number of events $\Delta N = \lambda$ | $k(T)$ per °C |
|---|---|---|
| -6.25 | 3 | 0.057 |
| -6.75 | 4 | 0.079 |
| -7.25 | 6 | 0.125 |
| -7.75 | 5 | 0.111 |
| -8.25 | 9 | 0.216 |
| -8.75 | 5 | 0.131 |
| -9.25 | 4 | 0.111 |
| -9.75 | 6 | 0.179 |
| -10.25 | 3 | 0.096 |
| -10.75 | 2 | 0.067 |
| -11.25 | 2 | 0.069 |
| -11.75 | 1 | 0.035 |
| -12.25 | 2 | 0.073 |
| -12.75 | 6 | 0.236 |
| -13.25 | 1 | 0.042 |
| -13.75 | 9 | 0.425 |
| -14.25 | 12 | 0.759 |
| -14.75 | 9 | 0.850 |
| -15.25 | 13 | 2.894 |

expected in repetitions. As can be seen for this example, it clearly isn't justified to attach too much significance to fine details of the spectrum, but there is reasonably good definition of the broad peak of activity centered on $-8$°C and of the rapid rise in numbers below $-12$°C. Should the observed data have been binned in larger temperature intervals, the confidence limits would have become narrower at the cost of lower temperature resolution. In the case here presented, this would be a reasonable choice even though the intuitive approach is to present the data with temperature resolution justified by measurement precision. The main limitation is from sample size.

As can be expected, the cumulative spectra are less sensitive to random variations in the number of freezing events per temperature interval. To illustrate this point, $K(T)$ is plotted for the 100 simulations in Fig. 9. Spread here reduces going toward lower temperatures and as values for more and more intervals are summed up. While at $-6.25$°C there is a factpr 10 spread in values, near $-15$°C the spread is about a factor of 2. This magnitude of error is for a sample of only 103 drops which is encouraging for experiments where larger drop numbers are not practical. Larger sample sizes can yield lower error ranges, but because the slope of the spectrum also has an influence no general statements are possible.

As an illustration of the influence of sample size on the confidence intervals for $k(T)$, the Snomax sample for which data were presented in Section 4 was also used in a Monte Carlo simulation. The input to the simulation was extracted from Table 1 for the region near the peak, where there are 30-50 events per bin. The simulation results for 100 iterations are shown in Fig.

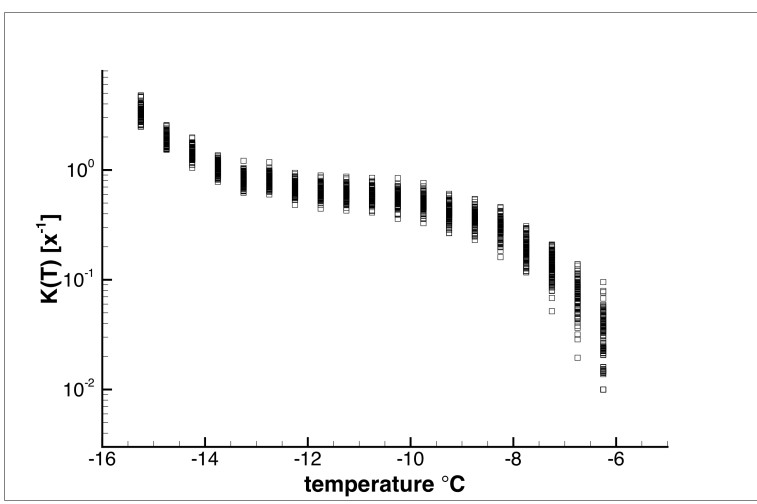

**Figure 9.** Cumulative spectra for 100 simulations for which differential spectra are shown in Fig. 7.

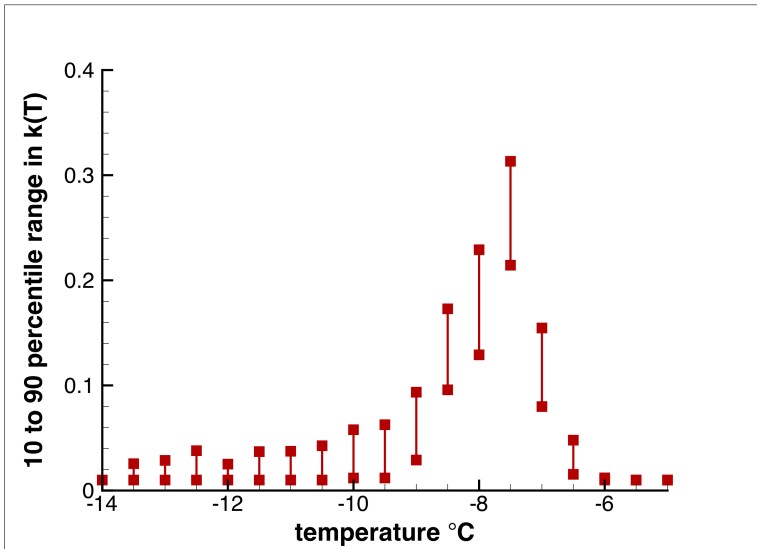

**Figure 10.** The 10 to 90 percentile range of $k(T)$ in 100 simulation for a segment of the spectrum shown in Fig. 4. Points just above the abscissa stand for zero values. In contrast with other figures, a linear ordinate scale is used because of the small range of values covered. A value of $X = 1$ is used; actual drop volume of particle concentration is not accounted for.

10 and, as can be seen, the range of variation is less than a factor 2 at the peak. At the lower $k(T)$ values, the variability is similar to what is seen in Fig. 10. Here too, zero values are plotted along an ordinate value of $10^{-2}$.

The examples shown above illustrate one possible way to assess the confidence limits of $k(T)$. The simulation approach is a realistic and readily envisioned method. Similar results for the confidence ranges could be obtained from tables of Poisson

distribution using the observed number of events in some experiment as $\lambda$ for each temperature interval. The standard deviation, $\lambda^{0.5}$, is another way to measure variability. However, it cannot be used in the way it would be for normally distributed values because, for example, the lower limit of the 95% range at $(\lambda - 2.14 * \lambda^{0.5})$ can be negative for small $\lambda$ and therefore not a physically realistic value for expected $\delta N$. The main point is that confidence limits can be delineated and with that the

5 meaning of derived $k(T)$ spectra quantitatively assessed. The results shown here also demonstrate the need for large sample sizes in order to reduce the variability of the derived spectra.

Once sample variability has been estimated, statistical methods are available for comparisons of two samples by testing, for example, the equivalence of means (e.g. Ch 20 in Blank (1980)). Performing that type of test interval by interval, as done in the foregoing, would test for activity in specific temperature regions. That may indeed be very useful in certain cases but will

definitely require large sample sizes. More complex methods will need to be considered to make broader overall comparisons of different samples. Combining data from larger temperature segments – those of greatest interest – could be helpful, but the strong temperature dependence of activity may be difficult to weigh adequately. Again, sample sizes will likely pose the most serious limitation to reaching statistical significance in such tests.

## 8   Active site density

Site density is defined in V15 as "The number of sites causing nucleation per unit surface area of the INP, or equivalent, as functions of temperature or supersaturation; the quantitative measure of the abundance of sites of different ice nucleating effectiveness.". Frequently, for added emphasis, the term is given as *active* site density and denoted as $n_s$. This quantity has already seen extended use in the literature (e.g. Connolly et al. 2009; Niedermeier et al. 2015; Beydoun et al. 2016; Paramonov et al. 2018; Boose et al. 2019). As stated in Section 2, normalization of the cumulative spectrum by particle surface area, using

$X = A$ in Eq. 4, leads to $n_s$, most frequently in the inverted form:

$$f(T) = 1 - exp(-A * n_s(T)). \tag{7}$$

No use has been made in the literature of the concept of differential active site density, although that metric has equal validity as the cumulative one, and is readily derived from Eq. 2 with the substitution of $X = A$.

Somewhat unfortunately, the active site density term was introduced in the literature in the cumulative form, i.e. activity

summed over all temperatures up to the test value. This happened because activity was generally understood to mean what is more precisely defined as the cumulative activity. The distinction between cumulative and differential activity is less widely appreciated. Following the general definitions of the differential and cumulative spectra, $k(T)$ and $K(T)$, it is useful to define differential and cumulative site density functions $k_s(T)$ and $K_s(T)$ recognizing that $K_s(T)$ is exactly equivalent to $n_s(T)$. If it weren't for the already established practice one could use the symbols $n_s(T)$ and $N_s(T)$, but it seems better to avoid the

confusion that could result when comparing results from different publications.

The two expressions for active site density are:

$$k_s(T) = -\frac{1}{A * \Delta T} * ln(1 - \frac{\Delta N}{N(T)}) \tag{8}$$

$$K_s(T) = -\frac{1}{A} * ln[1 - f(T)]. \tag{9}$$

Use of $A$ as average INP surface area included in each drop implies some important constraint on when that use is justified. First of all, it implies that the particles are stable and that the determination of $A$ was carried out in the suspension, not in the dry state. The two determinations may differ, for example, if the particles contain some soluble material, or they take up water and change in volume. Examples of aging and other effects altering particle effectiveness in water have already been reported (e.g. Emersic et al. 2015).Calculations of $k_s$ or $K_s$ for macromolecule INPs is of questionable value; these materials are best characterized with reference to the total mass of material or the number of individual macromolecules in suspension, not with reference to surface area. As for all of the quantitative characterizations discussed in this paper, temporal stability is assumed, at the minimum on the time scale of the experiment.

In addition to the considerations of the previous paragraph, valid use of an average surface area $A$ also requires that deviations from the mean value be reasonably small and not be the dominant source of error in the derived measures of activity. Special attention is needed with respect to the larger particles in polydisperse samples as these contribute disproportionate fractions of the total surface area. With sufficient knowledge of the particle size distributions error estimated can be derived for deviations from the average. Since $A$ appears in the pre-factor in the equations for both $k_s(T)$ and $K_s(T)$ the derived error estimate is valid for all values of the spectrum.

Dependent on the material constituting the INPs, total surface area may be an inadequate parameter to use in the calculation of the active site density. For example, if only a certain crystal face contains ice nucleating sites the surface area of that face is the relevant measure to include. Knowledge of such morphological factors is the goal of many studies; obtaining $k_s(T)$ or $K_s(T)$ with variations in experimental parameters may provide useful insights. On the other hand, without sufficient knowledge about particle surface characteristics substantial caveats need to be recognized regarding active site density spectra.

## 9 Summary

The differential spectrum, $k(T)$, is a useful representation of INP activity in heterogeneous freezing. This article examined some of the factors that need to be considered in derivations of $k(T)$ for experiments executed with gradual cooling of an array of sample drops taken from the same bulk sample, and with the freezing of drops at different temperatures recorded. Freezing at a given temperature is taken to indicate the presence of INPs active at that temperature. In Section 4, the importance of

**Table 4.** Nomenclature.

| | |
|---|---|
| $A$ | Average particle surface area contained in drops; $\text{m}^{-2}$ |
| $f(T)$ | Fraction of sample drops frozen at $T$ |
| $k(T)$ | Differential nucleus concentration; $\text{x}^{-1}\,^{\circ}\text{C}^{-1}$ |
| $k_s(T)$ | Differential active site density; $\text{m}^{-2}\,^{\circ}\text{C}^{-1}$ |
| $K(T)$ | Cumulative concentration of INPs active at temperatures above $T$; $\text{x}^{-1}$ |
| $K_s(T)$ | Cumulative site density on the INPs active at temperatures above $T$; $\text{m}^{-2}$ |
| $n_s(T)$ | Same as $K_s(T)$ |
| $N(T)$ | Number of drops not frozen at temperature $T$ |
| $\Delta N$ | Number of freezing events per temperature interval |
| $N_o$ | Total number of sample drops |
| $T$ | Temperature in $^{\circ}\text{C}$ |
| $T_i$ | Freezing temperature of a drop |
| $X$ | Reference quantity for normalization to unit volume of water, particle surface area, etc., as the case may be. For generality, corresponding units are indicated in $k(T)$ and $K(T)$ as x |
| $\lambda$ | Mean value of Poisson distribution, in the current context $\lambda = \Delta N_{observed}$ |

the choice of temperature interval for computing the spectra was elaborated. Methods of calculation and the relation to other derived quantities were presented in Section 5. Two applications were discussed: Section 6 presents a method for correcting empirical results for background effects. Correction for background is achieved by subtraction of the $k(T)$-values. In Section 7, a method was described for determination of confidence limits for $k(T)$ using Monte Carlo simulations. Sample size and spectral shape determine the error ranges of $k(T)$. Lesser uncertainty is associated with the cumulative spectra. The background correction and the determination of error ranges can significantly augment the value of information derived from laboratory freezing experiments and can improve model predictions of ice formation in clouds.

## 10   Data availability

Raw data of observed freezing temperatures for the three samples included in this paper are archived at the University of Wyoming under https://doi.org/10.15786/y5xr-pw35.

## 11   Acknowledgements

Dr. Ryan Sullivan of Carnegie Mellon University is thanked for permission to use the data given in Table 1. Thanks to Associate Editor Wiebke Frey for identifying errors in the original submission. Two anonymous referees made suggestions that were helpful for rounding out some of the arguments. Their questsions led to the clarification of some details.

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
