# Peer review of "Revisiting the differential freezing nucleus spectra derived from drop freezing experiments; methods of calculation, applications and confidence limits"

_Atmospheric Measurement Techniques, 2018_

## Referee Comment (RC1) · Anonymous Referee #2 · 6 Jan 2019

The derivation of the differential ice nucleus concentration metric ($k(T)$) is a valuable analysis method for droplet freezing assays commonly used to assess the concentration temperature spectrum of INPs. Droplet freezing assays have become widely used over the last few years, and are largely analyzed using the cumulative INP temperature spectrum ($K(T)$) also originally derived by Vali 1971. The differential form focused on here has some nice advantages compared to the common cumulative form, such as providing a more robust way to separate the contribution from background freezing interferences, and could see wide applicability throughout the growing ice nucleation

community. The differential form is not straight-forward to employ however, and the insights provided here will be of great value to those that wish to perform this analysis. I suspect that these issues are likely a cause for the lack of adoption of the k(T) metric by the ice nucleation community. Vali's efforts to clarify the use and utility of this metric will go a long way to help make this sort of quantitative analysis more widely practiced. The manuscript is well-written and easy to follow overall. I have a few comments and suggestions for revisions that might further improve the paper, which is certainly well within the scope of AMT.

This may be beyond the intent of this manuscript, but it seems that a very useful discussion regarding the important role that droplet size/volume and particle concentration play in accurately retrieving both K(T) and k(T) spectra and thus the INP(T) concentration spectrum would be extremely valuable to the ice nucleation community. Might these factors play a role in the reported discrepancies in n_s/K(T) from different methods of the same, as proposed by Emersic et al. (2015) and Beydoun et al. (2016)?

Page 2, line 2: These spectra also provide measurements of INP concentrations versus T.

Page 2, line 15: It would be good to also discuss the roles that droplet size and particle concentration play in the ability to retrieve information on INPs active at colder Ts.

3/7: This is an interesting and important point that I think is often overlooked by the growing number of groups using droplet freezing assays. It would be useful to elaborate more or this and further justify the notion that the approximation of one INP per droplet being responsible for the freezing event is valid when small delta-T values and large values of N are used. Please cite any key references here so more information on this can be readily found.

3/12: 10% error in what quantity? The approximation below in Eqn. 2?

4/20: Should mention that delta T is not just the temperature interval between successive droplet images, as might be commonly and erroneously thought. It appears to be a free parameter that must be adjusted for each dataset, as discussed further in the manuscript.

Section 7: It seems that a common way in which droplet freezing data are analyzed is to compare the freezing properties of one particle sample with others and determine if they are similar or different, or if some sort of processing or aging (heat, H2O2, acid, etc.) changes the freezing properties of the sample in a distinguishable manner. The use of confidence intervals applied to the k(T) differential analysis would seem to provide a quantitative way to determine if one particle sample distributed amongst an array or droplets has a freezing spectrum that is statistically similar or different from another. It could also be used to determine if data is significantly above the background freezing noise of the system. Could you please add a paragraph or more of discussion on these topics to this section? I think it will help to highlight the many different important types of analysis that can be done using these methods, and the utility of the derivative k(T) metric.

There are numerous little typos and misplaced words throughout the manuscript. Some of these may have been corrected since the original submitted abstract. I did not list them here but please proofread carefully to catch all of these.

Cited References

Beydoun, H., Polen, M. and Sullivan, R. C.: Effect of particle surface area on ice active site densities retrieved from droplet freezing spectra, Atmos. Chem. Phys., 16(20), 13359–13378, doi:10.5194/acp-16-13359-2016, 2016.

Emersic, C., Connolly, P. J., Boult, S., Campana, M. and Li, Z.: Investigating the discrepancy between wet-suspension- and dry-dispersion-derived ice nucleation efficiency of mineral particles, Atmos. Chem. Phys., 15(19), 11311–11326, doi:10.5194/acp-15-11311-2015, 2015.

---

## Referee Comment (RC2) · Anonymous Referee #1 · 20 Jan 2019

This is a well-written manuscript. The topic of the evaluation of different freezing nucleus spectra fits perfectly well into the journal AMT. The author revisits his own publications from 1971, 2008, and 2014. He develops enhanced methods of calculation, application, and confidence levels of freezing spectra from drop freezing experiments. The paper is very timely and the whole community of ice nucleation research will benefit from this paper which should be published after some minor corrections.

At the beginning of the paper the author should discuss in more detail the advantages to plot nucleation spectra as k(T) or K(T) functions and might compare with alternative

concepts. Many colleagues use ns, which is the site density. However, the cumulative concentration has many advantages over ns, which might be discussed. In particular, ns requires the specific surface area, which, e.g. in the case of soluble, macromolecular INP cannot be determined but requires suspendable, solid INP. Also, the specific surface area determined in gas phase can be rather different from ns in aqueous phase. Also in the cumulative functions there can be an impacts, when the normalization factor X is chosen, which can be normalization to unit volume, unit mass or unit surface of INP. The latter should be clarified and examples could be given, such as cellulose, which is even changing its freezing behavior from one freeze-thawing cycle to another, probably related with a change of specific surface area (water-cellulose interface).

Another interesting topic is the background correction. In figure 6, in the corrected differential spectrum a signal between -10°C and -17°C arises, which is probably related to the biological material in soil. This should be discussed in more detail, because it underlines the value and the significance of the whole method.

Minor corrections

P1, l15 and p2, l26: INP has already been defined in p1, l10 and doesn't need definition again.

P3, l1-2: "In practice, several runs with the same sample may be combined to accumulate a sufficiently large sample size No for useful statistical validity of the results." Add explanation. Repeated measurements might cause problems (see cellulose, comments above).

P4, l15: "Eqs. (1) of (2)" should be "Eq. (1) instead of (2)"

Fig. 3: The label of the left y-axis should be blue

Fig. 4: The label of the y-axis is not clear, use color code.

Fig. 5: Add "distilled water" to the label of the "control" curve

Fig. 6: Add "distilled water" to the label of the "control" curve

[Figure]

---

## Author Comment (AC1) · 3 Feb 2019

**Replies to Anonymous Referees #1 and #2 on "Revisiting the differential freezing nucleus spectra derived from drop freezing experiments; methods of calculation, applications and confidence limits."**

Gabor Vali

Department of Atmospheric Science, University of Wyoming, Laramie, Wyoming, USA

vali@uwyo.edu

February 1, 2019

**1   General**

Many thanks to both reviewers for their careful consideration of the paper. Their support for the aim and presentation of the paper is much appreciated. The suggestions they made are valuable and will improve the paper.

5       In slightly different ways, both referees ask for some discussion of the determination and application of active site density, $n_s(T)$[1], compared with $K(T)$. This is a valid point, as many recent publications have included use of this metric to evaluate results. The correspondence between $n_s(T)$ and the cumulative spectrum is already mentioned in the last paragraph of Section 2 of the first version of the paper. There has been no use made so far in the literature of the site density metric that corresponds to the differential concentration. Thus, impulsed by the referee comments, the revised manuscript has a brief section added

10     to discuss this, and to mention some of the factors that need to be considered for valid application of active site density as a measure of INP potency.

Other comments and suggested changes are taken up individually, as listed below.

In addition to the changes prompted by the reviews, I added to the manuscript a short paragraph and a new figure to show that the spread in the cumulative spectra from the simulation is much less than for the differntial spectrum. This is useful

15     to illustrate that even with modest numbers of drops error range of only about factors of 3-5 can be expected. The figure in question is reproduced here.

**2   Referee #1 comments**

*Referee:* "... ns requires the specific surface area, which, e.g. in the case of soluble, macromolecular INP cannot be determined."

Also in the cumulative functions there can be an impact, when the normalization factor X is chosen, which can be normalization
* * *
[1]Symbols are defined in Table 4 of the manuscript

[Figure]

**Figure 1.** Cumulative spectra for 100 simulations for which differential spectra are shown in Fig. 7.

to unit volume, unit mass or unit surface of INP. The latter should be clarified and examples could be given, such as cellulose, which is even changing its freezing behavior from one freeze-thawing cycle to another, probably related with a change of specific surface area (water-cellulose interface)."

*Response:* I have attempted to clarify in the new section under what conditions can $n_s$ be validly determined. Experiments with cellulose INPs raise additional issues regarding the stability of the particle surface once interaction with water starts. I have no experience with this system, so cannot add more specific recommendations beyond those included in the preceding section. Actually, this is just one of the many systems which undergo changes with time and for which $n_s$ or any other measure of activity is a function of time, Changes even during the course of a single experiment can occur and make any quantitative description of the activity of questionable value.

*Referee:* "In figure 6, in the corrected differential spectrum a signal between -10°C and -17°C arises, which is probably related to the biological material in soil."

*Response:* The origin of the INPs active at the higher temperatures in that analysis were not determined. The referee's view is probably correct as very few materials are known to produce nucleation at those temperatures. However, since no independent determination was made of the INP composition, a claim of biological origin has only inferential support. Also, it is a coincidence that in the example chosen to demonstrate the background subtraction there is a well-defined peak in the differential spectrum and the minimum on the left of this peak occurs just where the subtraction becomes significant. This is not expected to be the general case. I'll mention this in the revised manuscript.

*Referee:* "P3, l.1-2: In practice, several runs with the same sample may be combined to accumulate a sufficiently large sample size $N_o$ for useful statistical validity of the results.? Add explanation. Repeated measurements might cause problems (see cellulose, comments above)."

*Response:* Correct. Combination of several runs is not an option if the sample is unstable. Text to emphasize this has been added to the manuscript.

*Referee:*"P4, l.15: "Eqs. (1) of (2)" should be "Eq. (1) instead of (2)" "
*Response:*Meant to say Eqs. (1) **or** (2), since the question of choice of $\Delta(T)$ is relevant to both expressions.

**3 Referee #2 comments**

*Referee:* "Page 2, line 2: These spectra also provide measurements of INP concentrations versus T."
*Response:* Thanks. Made that sentence more complete.

*Referee:* "Page 2, line 15: It would be good to also discuss the roles that droplet size and particle concentration play in the ability to retrieve information on INPs active at colder Ts."
*Response:* Added a paragraph about this. "From and experimental perspective, quantitation of ice nucleating ability depends on a successful choice of the drop sizes and of the amount of suspended INPs. Because ice nucleating ability in general is a strong function of temperature, small drop volumes and low amounts of particle content result in freezing temperatures at low temperatures. On the contrary, with large drop volumes and high particle loading, most drops will freeze at roughly the same temperature. The range of usable drop volumes is often defined by the design of the apparatus, but, for laboratory preparations, particle concentration is controlled by the experimenter. For water samples obtained with indigenous INPs (rain, river water, etc.) particle concentrations can be altered by dilution and partial evaporation. The functions defined in the following section are useful only when the data to be analyzed describe a substantial spread of observed freezing temperatures."

*Referee:* "3/7: This is an interesting and important point that I think is often overlooked by the growing number of groups using droplet freezing assays. It would be useful to elaborate more or this and further justify the notion that the approximation of one INP per droplet being responsible for the freezing event is valid when small delta-T values and large values of N are used. Please cite any key references here so more information on this can be readily found." and "3/12: 10% error in what quantity?"
*Response:* The difference between containing at least one active INP or having exactly one is indeed easily overlooked. The higher the average concentration is the less likely it is that precisely one INP would be found per drop. The statements made in the cited paragraph derive from the properties of the Poisson distribution. The derivations given in V71 account for this. The reference was added.

*Referee:* "4/20: Should mention that delta T is not just the temperature interval between successive droplet images, as might be commonly and erroneously thought. It appears to be a free parameter that must be adjusted for each dataset, as discussed further in the manuscript."
*Response:* Indeed, this is a point of potential source of errors worth mentioning along with the other unadvisable choices for $\Delta T$ given in Section 4 of the paper. Added the sentence:"The choice is made, principally, on the basis of sample size (number of drops in the experiment) and not based on instrumental variables, such as the recording interval of freezing events."

*Referee:* "Section 7: It seems ... common ... to compare the freezing properties of one particle sample with others ... The use of confidence intervals applied to the k(T) differential analysis would seem to provide a quantitative way to determine if one particle sample distributed amongst an array or droplets has a freezing spectrum that is statistically similar or different from another. ... please add a paragraph or more of discussion on these topics ... "

5     *Response:* Thanks for the suggestion. The following text was added in the revised manuscript: "The examples shown above

illustrate one possible way to assess the confidence limits of $k(T)$. The simulation approach is a realistic and readily envisioned method. Similar results for the confidence ranges could be obtained from tables of Poisson distribution using the observed number of events in some experiment as $\lambda$ for each temperature interval. The standard deviation, $\lambda^{0.5}$, is another way to measure variability. However, it cannot be used in the way it would be for normally distributed values because, for example,

10    the lower limit of the 95% range at $(\lambda - 2.14 * \lambda^{0.5})$ can be negative for small $\lambda$ and therefore not a physically realistic value for expected $\delta N$. The main point is that confidence limits can be delineated and with that the meaning of derived $k(T)$ spectra quantitatively assessed. The results shown here also demonstrate the need for large sample sizes in order to reduce the variability of the derived spectra. "

"Once sample variability has been estimated, statistical methods are available for comparisons of two samples by testing,

15    for example, the equivalence of means (e.g. Ch 20 in Blank (1980)). Performing that type of test interval by interval, as done in the foregoing, would test for activity in specific temperature regions. That may indeed be very useful in certain cases but will definitely require large sample sizes. More complex methods will need to be considered to make broader overall comparisons of different samples. Combining data from larger temperature segments – those of greatest interest – could be helpful, but the strong temperature dependence of activity may be difficult to weigh adequately. Again, sample sizes will likely pose the most

20    serious limitation to reaching statistical significance in such tests."

**4   Corrections**

Minor corrections in figure labels and in wording will be made as suggested by the referees.